# Prevalence and Persistence of Symptoms in Adult COVID-19 Survivors 3 and 18 Months after Discharge from Hospital or Corona Hotels

**DOI:** 10.3390/jcm11247413

**Published:** 2022-12-14

**Authors:** George Kalak, Amir Jarjou’i, Abraham Bohadana, Pascal Wild, Ariel Rokach, Noa Amiad, Nader Abdelrahman, Nissim Arish, Chen Chen-Shuali, Gabriel Izbicki

**Affiliations:** 1Department of Medicine, Pulmonary Institute, Shaare Zedek Medical Center, Faculty of Medicine, Hebrew University of Jerusalem, Jerusalem 9112102, Israel; 2PW Statistical Consulting, 54520 Laxou, France; 3Faculty of Medicine, Hebrew University of Jerusalem, Jerusalem 9112102, Israel

**Keywords:** long COVID, COVID-19 symptoms, persistent symptoms

## Abstract

COVID-19 is characterized by persistent symptoms beyond acute illness. In this prospective cohort study of patients with COVID-19, we sought to characterize the prevalence and persistence of symptoms up to 18 months after diagnosis. We followed 166 patients and assessed their symptoms during acute illness, and at 3 and 18 months after disease onset. The mean number of symptoms per patient during acute disease was 2.3 (SD:1.2), dropping to 1.8 (SD:1.1) at 3 months after recovery and to 0.6 (SD:0.9) at 18 months after recovery. However, this decrease was not unidirectional. Between acute illness and 3 months, the frequency of symptoms decreased for cough (64.5%→24.7%), ageusia (21.7% to6%), anosmia (17.5%→5.4%), and generalized pain (10.8% to 5.4%) but increased for dyspnea (53%→57.2%) weakness (47%→54.8%), and brain fog (3%→8.4%). Between 3 and 18 months, the frequency of symptoms decreased for all symptoms but remained relatively high for dyspnea (15.8%), weakness (21.2%), and brain fog (7.3%). Symptoms may persist for at least 18 months after acute COVID-19 infection. During the medium- to long-term recovery period, the prevalence of some symptoms may decrease or remain stable, and the prevalence of others may increase before slowly decreasing thereafter. These data should be considered when planning post-acute care for these patients.

## 1. Introduction

Coronavirus-2019 (COVID-19) disease is caused by the novel virus called severe acute respiratory syndrome coronavirus 2 (SARS-CoV-2). In adults, the spectrum of COVID-19 manifests with a variety of clinical presentations and symptoms, ranging from asymptomatic infection to mild respiratory symptoms to severe acute respiratory distress syndrome [1]. While at the beginning of the pandemic, medical attention focused primarily on the acute aspects of the disease, there is now ample evidence that considerable morbidity persists beyond the acute phase, with a significant proportion of patients presenting with symptoms for long periods after recovery. “Long COVID” or “post-acute COVID-19” is characterized by persistent symptoms that prevent patients from returning to work and a normal life [2]. The exact prevalence of “post-acute COVID-19” is not yet known; however, given the large number of diagnosed cases, the numbers are likely to be high, even for low prevalence rates. In this context, follow-up studies of symptoms in survivors of acute infection are important to help plan post-acute care for these patients.

The heterogeneity of presenting symptoms is an additional reason to conduct follow-up studies in COVID-19 survivors. Indeed, for a given patient, symptoms depend on the interaction between virus–host factors, including viral load [3]. After binding to angiotensin-converting enzyme-2 (ACE-2) receptors on the surface of the nasal and upper respiratory tract, the virus spreads throughout the body and can affect a wide range of organ systems [1]. In this scenario, although dyspnea and cough can logically be expected to affect the majority of patients [2,4], these symptoms may be accompanied or even supplanted by others, including fatigue, muscle weakness, sleep disturbance, joint pain, myalgia’s, gastrointestinal symptoms, cognitive symptoms, and headaches [5,6,7,8,9,10].

Given the above considerations, we sought to characterize the prevalence and persistence of symptoms associated with COVID-19 infection in a cohort of patients seen in the medium (3 months) and long term (18 months) after discharge from hospital or Corona hotels. Our objective was to examine the temporal course of symptoms from onset to 18 months after hospital discharge.

## 2. Methods

### 2.1. Study Design and Participants

This prospective, single-center cohort study of patients with COVID-19 infection requiring hospitalization was conducted at Shaare Zedek Medical Center (SZMC) in Jerusalem, a leading center for the treatment of patients with COVID-19 in Israel. Patients aged 18 years or older, discharged from the hospital or Corona Hotels between June and November 2020, were invited to participate in a visit 3 months later. Corona Hotels are specific facilities designated by the Israeli Ministry of Health to accommodate patients diagnosed with mild to moderate COVID-19, providing appropriate, self-monitored medical care. Recovery is defined as being healthy enough to be discharged from the hospital or Corona Recovery Hotels and no longer need to be quarantined.

In all cases, the diagnosis of COVID-19 was based on real-time polymerase chain reaction (RT-PCR) of samples obtained by oropharyngeal or nasopharyngeal swabs [11]. Pediatric patients, pregnant women, bedridden patients, and patients with cognitive impairment were excluded from the study. The study was approved by the “Helsinki Ethics Committee” of SZMC [Reg: 0174-20-SZMC]. All participants signed a written informed consent.

### 2.2. Procedures

Data collection was performed at the COVID-19 post-acute clinic at the SZMC Lung Institute. Three months after recovery, eligible participants, after signing a written consent, underwent a standard clinical evaluation with assessment of symptoms, health status, pulmonary function tests, echocardiography, and a 6 min walk test. At the end of 3 months, during a face-to-face examination, a pulmonary physician asked the participants about their symptoms using an open-ended questionnaire and a list of the most common symptoms of COVID-19 infection in our clinical experience, namely cough, dyspnea, chest pain, weakness, loss of smell (anosmia), loss of taste (ageusia), generalized pain, and “brain fog,”, i.e., difficulty thinking clearly. All symptoms were listed without interpretation. Health-related quality of life was assessed at 3 months using the free-form version of the Short-form Health Survey [SF-36] questionnaire, which assesses eight physical and mental domains (see list in the eSupplement, Part I) [12]. For each participant, acute-phase clinical data, defined as the time from symptom onset to discharge from the hospital (or Corona Hotel), were extracted from electronic medical records. These data included initial signs and symptoms leading to hospitalization, self-reported comorbidities, and the Charlson Comorbidity Index (CCI) [13,14]. In addition, disease severity was defined according to the NIH COVID-19 treatment guidelines [15] as follows: (a) Mild disease: Individuals with any of the various signs and symptoms of COVID-19 (e.g., fever, cough, sore throat, malaise, headache, muscle aches, nausea, vomiting, diarrhea, loss of taste and smell) but without dyspnea or abnormal chest imaging; (b) Moderate disease: Individuals with evidence of lower airway disease on clinical or imaging evaluation and an oxygen saturation (SpO_2_) ≥ 94% on room air at sea level; (c) Severe disease: Individuals who have an SpO_2_ < 94% on room air at sea level, arterial partial pressure of oxygen to fraction of inspired oxygen (PaO_2_/FiO_2_) < 300 mm Hg, respiratory rate >30 breaths/min, or pulmonary infiltrates > 50%; and d) Critical disease: Individuals with respiratory failure, septic shock, and/or multi-organ dysfunction. [15]. Participants who attended the 3-month post-recovery visit were contacted by telephone by one of two authors (AJ and GK) 15 months later. Symptoms were then reassessed using the same open-ended questionnaire and the predefined list of symptoms described above.

## 3. Outcomes of the Study

The primary outcome was persistence of symptoms in patients infected with COVID-19.

## 4. Statistical Analysis

Statistical analysis was performed using the Stata software package (Stata 17, College Station, TX, USA). The eight symptoms (cough, dyspnea, chest pain, weakness, anosmia, ageusia, generalized pain, and brain fog) reported during the acute phase of COVID-19 and studied at 3 and 18 months after recovery were the primary outcomes analyzed. To summarize treatment and illness in a meaningful way, we generated two new variables: The first variable, named CatHospt, combines information on hospitalization, length of stay (cutoff point 1 week), and whether the patient was admitted to intensive care. The second, named TreatItems, combines information on oxygen use, high-flow nasal cannula (HFNC), and intubation. Next, the number of symptoms 3 and 18 months after recovery was modeled using pooled logistic regression with all variables recorded during acute COVID-19 (disease history, disease severity and treatment, and symptoms), baseline characteristics (age, sex, smoking history, Charleston comorbidity index, number of comorbidities, and each comorbidity as a separate variable), with each variable considered separately as an independent factor. Finally, selected symptoms that increased in prevalence between acute COVID-19 and 3 months after recovery or showed high prevalence at follow-up were analyzed by logistic regression considering the same variables recorded during acute COVID-19. A *p* value less than 0.05 was considered significant.

## 5. Results

Of the 203 patients who visited our COVID-19 post-acute clinic between June and November 2020, 166 agreed to participate in the study and were included. Of these, 111 (66.9%) had been treated at our hospital and 55 (33.1%) at Corona hotels around the country. A flow chart of the recruited patients is presented in Figure 1.

## 6. Baseline Characteristics

Table 1 presents the demographic and clinical characteristics of the participants. Overall, the cohort was middle-aged, with a mean age of 52.1 years (SD 16.8) and a male-to-female ratio of 50%. There were only five active smokers (3.0%) in the sample population. Four comorbidities-hypertension, diabetes, asthma, and obstructive sleep apnea (OSA)-accounted for 68% of reported cases (88/129).

Overall, the severity of comorbidities was mild, with a mean Charlson Comorbidity Index (CCI) of 1.6 (SD, 1.9). Regarding treatment, slightly more than one-third of patients required oxygen; of 15 (9%) patients admitted to intensive care, 9 were intubated. Of the 29 patients who required HFNC, only one was also treated with BIPAP for 3 days. No other noninvasive mechanical ventilation modalities were used. The most common treatments were enoxaparin sodium (Clexane), hydroxychloroquine, dexamethasone, remdesivir, and azithromycin.

## 7. Symptom Evaluation

The mean number of symptoms per patient at the time of acute illness was 2.3 (SD:1.2), falling to 1.8 (SD:1.1) 3 months after recovery and to 0.6 (SD:0.9) 18 months after recovery. Figure 2 shows the course of the number of symptoms per patient from diagnosis to 3 and 18 months after recovery. In acute COVID-19, 160 (96.4%) patients complained of ≥ 1 symptom, with 123 (73.5%) reporting ≥ 2 symptoms. At 3 months, symptom prevalence was still high, with 156 (95.8%) patients complaining of ≥ 1 symptom; however, there was a shift of the curve to the left, with fewer patients-102 (61.5%)-reporting ≥ 2 symptoms. At 18 months, the prevalence of symptoms decreased significantly, with 64 reporting at least 1 symptom and 22 (13.3%)-reporting ≥ 2 symptoms.

Figure 3 shows the prevalence of each symptom at different time points in the study. Fever, present in almost 100% of patients at disease onset, is not a symptom and was excluded from this analysis. The most common symptoms at disease onset were cough (64.5%), dyspnea (53%), and weakness (47%), followed by ageusia (21.7%), anosmia (17.5%), chest pain (15.0%), generalized pain (10.8%), and brain fog (3.0%). Between acute illness and 3 months after recovery, the frequency of symptoms decreased for cough (from 64.5% to 24.7%), ageusia (from 21.7% to 6%), anosmia (from 17.5% to 5.4%) and generalized pain (from 10, 8% to 5.4%), did not change significantly for chest pain (from 15% to 17.5%) but increased for dyspnea (from 53% to 57.2%), weakness (from 47% to 54.8%), and brain fog (from 3% to 8.4%). Finally, between 3 and 18 months after recovery, the frequency of symptoms fell to zero for ageusia, anosmia and chest pain, decreased markedly for cough (from 24% to 7.9%), remained unchanged for generalized pain (from 5.4% to 5.5%), and decreased to a relatively high value for dyspnea (from 57% to 15.8%), weakness (from 54.8% to 21.2%), and brain fog (from 8.4% to 7.3%).

### 7.1. Predictors of the Number of Symptoms

Using a pooled logistic model, no predictive effect was observed between demographic characteristics-including age, sex, BMI, and smoking habits-and the number of symptoms at either 3 or 18 months after recovery. Regarding history, among factors including the disease itself (with each comorbidity considered a separate variable), as well as CHF and number of medications, only obstructive sleep apnea (OSA) was predictive of number of symptoms at 3 months (*p* = 0.017). None of these factors were predictive of the number of symptoms at 18 months. With respect to treatment parameters, neither of the two summary variables characterizing disease and treatment, nor disease severity (based on the NIH severity index) as a separate variable, significantly predicted the number of symptoms 3 months after recovery.

However, the variable characterizing intensive treatment was significantly predictive (*p* = 0.014) of the number of symptoms 18 months after recovery (data not shown), with 1.2 symptoms among the 9 intubated patients (3 were still coughing) significantly higher than the 0.6 symptoms among the 102 patients without oxygen or HFNC, whereas the number of symptoms (0.25) was lower among the 20 patients treated with HFNC than among the 102 patients without oxygen or intubation. Finally, the number of symptoms at diagnosis was a marginally significant predictor of the number of symptoms three months after recovery (*p* = 0.04). Among the symptoms themselves, dyspnea (*p* = 0.006) and “brain fog” (*p* = 0.023) at diagnosis were predictive of the number of symptoms three months after recovery. The latter result, however, is based only on the five patients with this symptom and should not be over-interpreted. Regarding the number of symptoms 18 months after recovery, the number of symptoms at diagnosis was not a significant predictor (*p* = 0.78), but dyspnea at diagnosis significantly (*p* = 0.02) predicted the number of symptoms 18 months after recovery.

### 7.2. Symptoms and Health-Related Quality of Life

There was a decrease in the mean scores of the SF-36 subscales for the physical and mental components, primarily the Physical Role, Emotional Role, and Vitality subscales (Data presented in the eSupplement, Part II). Furthermore, all dimensions explored by the SF36 questionnaire were significantly correlated with the number of symptoms, with the highest correlation coefficients observed for Physical Functioning (r = 0.40), Physical Role Limitations (r = 0.39), and Body Pain (r = 0.36).

### 7.3. Predictors of Selected Symptoms

The following factors were significant predictors of dyspnea three months after recovery: obesity defined as BMI > 30 (Or = 2.27, *p* = 0.022), number of comorbidities (Or = 1.26, *p* = 0.014), oxygen support (Or = 1. 97, *p* = 0.041); treatment with enoxaparin sodium (Or = 4.63, *p* < 0.0001) or dexamethasone (Or = 2.46 *p* = 0.035); hypertension (2.28, *p* = 0.038) and dyspnea during acute COVID (Or = 3.25, *p* < 0.0001). However, it should be noted that after adjustment for the drug enoxaparin sodium, no other variables except dyspnea during acute COVID remained significant. At 18 months, men reported more dyspnea (Or = 2.3, *p* = 0.035). The only other factors were dexamethasone treatment (Or = 3.0, *p* = 0.017) and dyspnea at the time of acute VaDOC (Or = 3.58, *p* = 0.010). Weakness at the time of acute COVID-19 was the only variable predicting an effect on reported weakness at 3 months after recovery (*p* < 0.001). At 18 months, women reported more weakness (Or = 2.3, *p* = 0.035). Finally, “brain fog” (Or = 8.3, *p* = 0.028) at the time of acute COVID-19 was a predictor of “brain fog” three months after recovery. At 18 months after recovery, no other variables recorded at the time of acute COVID were predictive.

## 8. Discussion

This cohort study describes the long-term course of post-acute symptoms associated with COVID-19 infection in patients discharged from hospitals or Corona hotels in Israel. Demographics, comorbidities, and severity of illness at baseline were comparable to previous studies [7]. Overall, patients were middle-aged, with a mean age of 52.1 years. Consistent with previous studies, the cohort included an equal proportion of men and women [9,16] and a low prevalence of smokers [17]. Disease at diagnosis was mild to moderate in 53% of patients, severe in 40.4%, and critical in 6.6%; only 9 patients were intubated.

In our cohort, the most common symptoms at disease onset were, in decreasing order of prevalence, cough, dyspnea, weakness, ageusia, anosmia, chest pain, generalized pain, and brain fog. Interestingly, the progression of symptoms over time was not unidirectional. The prevalence of three symptoms—ageusia, anosmia, and chest pain—decreased steadily over time and disappeared 18 months after recovery. For their part, cough and generalized pain also decreased in prevalence, but at a slower rate, and were still reported 18 months after cure by a significant minority of patients. Somewhat intriguingly, dyspnea, weakness, and brain fog showed an increase in prevalence at 3 months, followed by a slowdown thereafter, so that 18 months after recovery, these symptoms were still reported by a relatively large proportion of patients compared with the onset of the disease. In this regard, although modest in absolute terms, the prevalence of brain fog more than doubled at 3 and 18 months after recovery. Therefore, the most common symptoms at disease onset differed from those observed at mid- and long-term follow-up. Finally, the low scores of the mental and physical components of the SF-36 questionnaire and their significant correlation with the number of symptoms are consistent with a significant alteration of the patients’ quality of life, which is important in the perspective of their rehabilitation process.

The fluctuating pattern of post-COVID symptoms has been noted previously. A systematic review and meta-analysis examining the prevalence of post-COVID-19 symptoms in individuals who recovered from COVID-19 [18] showed that post-COVID-19 symptoms experienced 30 days after illness onset or hospitalization decreased significantly in prevalence compared to the acute phase but increased 60 days after. The authors emphasized the need for longitudinal studies, as their prevalence data were based on a small number of studies and comparisons were very heterogeneous.

Using a longitudinal cohort design, Seeßle and colleagues [19] evaluated a sample of COVID-19 survivors at 5, 9, and 12 months after infection and found that between 5 and 12 months after symptom onset, the frequency of reported symptoms decreased significantly only for hair loss (from 26.1% to 10.4%) but increased for fatigue (from 41.7% to 53.1%) and dyspnea (from 27.1% to 37.5%). However, as the authors acknowledged, the results may have been influenced by the fact that 50 of the 146 patients who initially consented to participate in the study and were seen at 5 months were lost to follow-up at 12 months and were therefore excluded from the long-term analysis of the 12-month follow-up. In contrast, in our study, only 1 of 166 patients seen at the 3-month visit was lost to follow-up at the 18-month follow-up visit and was excluded from the analysis. Therefore, our data provide strong support for the concept that the fluctuating nature of post-COVID symptomatology is real and is not an artifact resulting from the pooling of data from patients remaining in the study.

Consistent with previous studies, we found that some factors were predictive of symptoms after recovery. For example, a history of OSA predicted symptoms 3 months after recovery, but not 18 months later. Similarly, dyspnea, weakness, and brain fog at diagnosis were predictors of the same symptoms at follow-up. Although interesting, these results are difficult to interpret and especially to compare with the literature. Indeed, these associations are likely to be influenced by a large number of variables such as, for example, sample size, age, gender distribution, social habits, comorbidities and health status of the participants, as well as the severity of the disease and the methods used to assess the symptoms and associated parameters.

Therefore, the predictive factors are very heterogeneous. To cite just a few examples, previous studies have reported groups of factors associated with long COVID as varied as (a) experience of more than 5 symptoms in the first week [20]; (b) age 40–60 years, hospital admission, and abnormal auscultation at the time of symptom onset [21,22]; (c) lower rates of SARS-CoV-2-IgG, anosmia, and diarrhea during acute illness [23]; and (d) female gender, number of comorbidities, having more than 5 symptoms on hospital admission, and number of days in hospital [24,25,26]. In this context, an in-depth analysis of this topic is beyond the scope of our study. Notably, the increase in symptoms at 3 months cannot be attributed to reinfection, as all our participants suffered from COVID-19 in the first waves in 2020. Reinfection within 3 months was very rare at that time and the screening rate was very high. In our study, we reviewed the patients’ electronic medical records and found no evidence of reinfection in the first 3 months after recovery in any of them.

Consistent with previous reports [17], the prevalence of smoking in our cohort was quite low. Because ACE-2 receptors are known to be a hallmark of tissue vulnerability to infection, the previously described upregulation of the ACE-2 gene in smokers compared with nonsmokers [27,28,29,30,31,32] would imply that smokers have an increased risk of SARS-CoV-2 infection. If this were true, the low prevalence of smoking in our cohort could represent the “survival of the fittest” phenomenon; however, because we do not have data on smoking in patients who died of infection, we can only speculate on this question.

One of the strengths of our study is the long-term follow-up of the entire study sample, with only one patient lost to follow-up between the 3- and 18-month follow-up visits. As for limitations, the single-center, unblinded, uncontrolled study design and relatively small cohort size may limit the generalizability of the results. Second, we lack information on the health status of patients in the months before infection with the virus, which limits our ability to conclude with certainty that follow-up outcomes were temporally related to COVID-19. Finally, the severity of reported symptoms was not assessed; however, symptoms were assessed by physicians in a structured manner, so we can assume that no important symptoms were missed.

One could argue that this study is limited by the lack of information on four important variables that theoretically could have influenced patients’ symptoms, namely vaccination, reinfection, high-dose steroid use, and long COVID variants. However, our patients were seen during the first wave of COVID, when no vaccine was available, no COVID-19 variants had been described, and no guidelines for treatment of long COVID existed. Although reinfection and vaccination status could not explain the increase in symptoms at 3 months, we did not test for reinfection or vaccination status at 18 months and cannot predict the effect of reinfection or vaccination on symptoms at 18 months. In addition, treatment recommendations for acute COVID-19, including steroid use, were evolving at the time; moreover, we used this treatment in 20.5% of subjects (Table 1).

In conclusion, in adult survivors of COVID-19 infection, symptoms may persist for at least 18 months after recovery and significantly impair quality of life. Our results confirm the fluctuating pattern of post-COVID-19 symptoms over time; the prevalence of some symptoms decreasing or remaining stable and that of others increasing in the medium and long term. These data provide useful information for the clinical management of patients recovering from COVID-19. Longer follow-up studies in larger populations are needed to fully characterize the time course of symptoms in COVID-19 survivors.

## Figures and Tables

**Figure 1 jcm-11-07413-f001:**
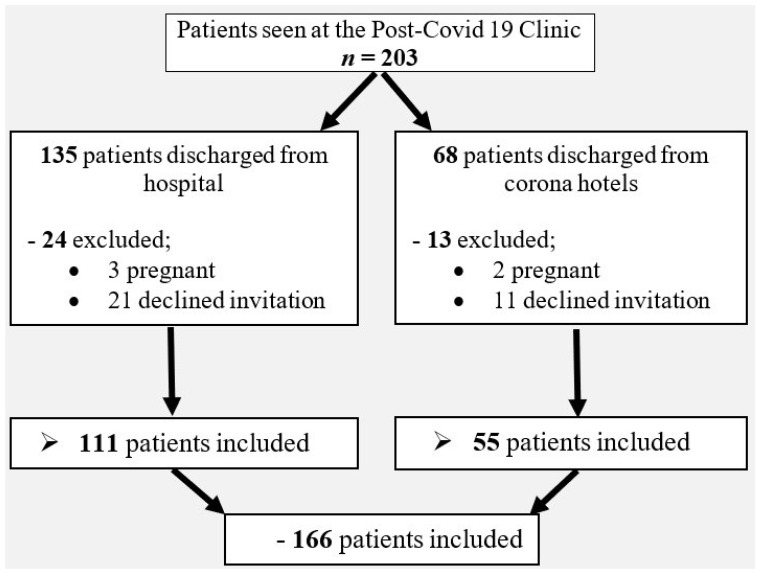
Flow-Chart of Survivors to COVID infection participating in the study. Of the 203 patients who attended our post-COVID clinic between June and November 2020, 166 patients were included in our follow-up study. 111 patients were discharged from our hospital, 55 patients were discharged from recovery Corona hotels. Of the 37 patients who were not included, 32 declined to participate and 5 were pregnant and therefore not included.

**Figure 2 jcm-11-07413-f002:**
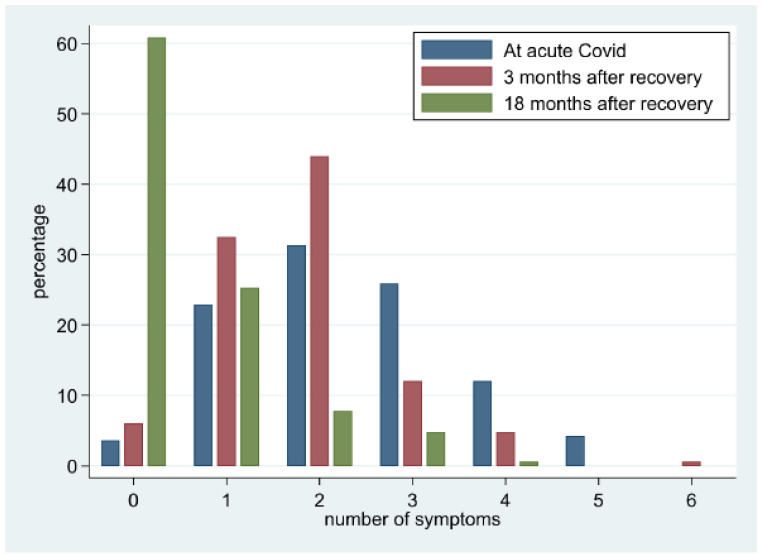
Proportion of COVID survivors stratified by the number of symptoms at acute disease and 3 and 18 months after recovery. This figure shows the percentage of COVID-19 survivors (Y axis) stratified by the number of symptoms (X axis) at baseline (blue), 3 months (red), and 18 months (green). At acute COVID-19, 160 (96.4%) patients complained of ≥ 1 symptom, with 123 (73.5%) reporting ≥ 2 symptoms. At 3 months, the prevalence of symptoms was still high, with 156 (95.8%) patients complaining of ≥ 1 symptom; however, there was a shift of the curve to the left, with fewer patients-102 (61.5%)-reporting ≥ 2 symptoms. At 18 months, the prevalence of symptoms decreased significantly, with 64 reporting at least 1 symptom and 22 (13.3%)—reporting ≥ 2 symptoms.

**Figure 3 jcm-11-07413-f003:**
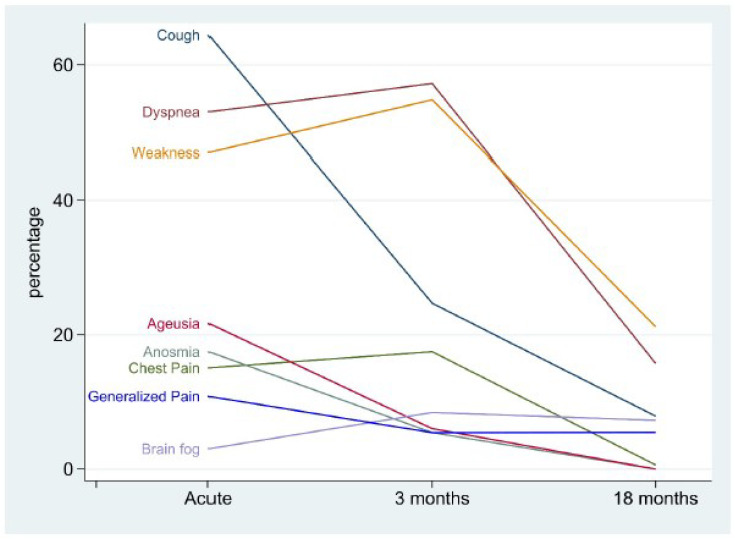
Fluctuation of symptoms from onset of the disease through 3 and 18 months post-recovery in COVID-19 survivors. This figure shows the prevalence of each symptom (Y-axis) across the three study time points (X-axis), at the time of acute illness, at 3 months and at 18 months. At the onset of illness, the most common symptoms were cough (64.5%), dyspnea (53%), and weakness (47%), followed by ageusia (21.7%), anosmia (17.5%), chest pain (15.0%), generalized pain (10.8%), and brain fog (3.0%). At 3 months, the most common symptoms were dyspnea (57.2%), weakness (54.8%), and cough (24.7%), followed by chest pain (17.5%), brain fog (8.4%), age (6%), anosmia (5.4%), and generalized pain (5.4%). At 18 months, the most common symptoms were weakness (21.2%) and dyspnea (15.8%), followed by cough (7.9%), brain fog (7.3%), and generalized pain (5.5%). The frequency of chest pain at 18 months was 0.6%, and zero for anosmia and ageusia.

**Table 1 jcm-11-07413-t001:** Characteristics of enrolled patients.

Parameters	*n* = 166
**Age, yrs. (mean, SD)**	52.1 ± 16.8
Age, yrs. (range)	19–86
**Male *n* (%)**	83 (50)
**Female *n* (%)**	83 (50)
**BMI kg/m^2^ (mean, SD)**	28.1 ± 5.8
**Smoking Status**	
Current smokers *n* (%)	5 (3.0)
Former smokers *n* (%)	25 (15.1)
Never smokers *n* (%)	136 (81.9)
**Comorbidities (mean, SD)**	1.8 ± 2.1
Hypertension *n* (%)	39 (23.5)
Diabetes *n* (%)	28 (16.9)
Asthma *n* (%)	11 (6.6)
OSA * *n* (%)	10 (6.0)
Ischemic Heart Disease *n* (%)	9 (5.4)
Connective tissue disease *n* (%)	8 (4.8)
Chronic Heart Failure *n* (%)	6 (3.6)
Chronic Renal Failure *n* (%)	4 (2.4)
COPD * *n* (%)	3 (1.8)
Immunosuppressive disease *n* (%)	3 (3.8)
Solid Tumor *n* (%)	2 (1.2)
Lymphoma *n* (%)	1 (0.6)
ILD * *n* (%)	1 (0.6)
Other respiratory *n* (%)	4 (2.4)
**Charlson Comorbidity Index (mean, SD)**	1.6 ± 1.9
**Disease Severity NIH scale**	
Mild	51 (30.7)
Moderate	37 (22.3)
Severe	67 (40.4)
Critical	11 (6.6)
**Treatment**	
**Oxygen *n* (%)**	64 (38.6)
**HFNC *n* (%)** *^$^	29 (17.5)
Days on HFNC	2.24 ± 3.7
**ICU admission *n* (%)**	15 (9.0)
**Intubation** ***n* (%)**	9 (5.4)
Duration in days (mean, SD)	0.9 ± 4.0
**Drug Treatment**	
Hydroxychloroquine *n* (%)	39 (23.5)
Azithromycin *n* (%)	27 (16.3)
Remdesivir *n* (%)	15 (9.0)
Dexamethasone *n* (%)	34 (20.5)
Enoxaparin sodium *n* (%)	83 (50.0)
Other antibiotics *n* (%)	19 (11.5)

* OSA: Obstructive Sleep Apnea, COPD: Chronic Obstructive Lung Disease, ILD: Interstitial Lung Disease, HFNC: High flow nasal cannula. ^$^ Of the 29 patients on HFNC, one patient was also treated with BIPAP, no other modality of NIMV was used.

## Data Availability

All participant-level data relevant to the study are included in the article. The study data are available from the corresponding author upon reasonable request, after removal of all personal identifiers, and after approval by the SZMC ethics committee.

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
