# Peer review of "Prevalence and Persistence of Symptoms in Adult COVID-19 Survivors 3 and 18 Months after Discharge from Hospital or Corona Hotels"

_jcm, 2022, doi:10.3390/jcm11247413_

Round 1

Reviewer 1 Report

This paper presents the results of an interesting study about the fluctuating pattern of COVID-19 in 166 patients over a time period of 18 months. However, I have several major concerns. Furthermore, the paper can be shortened considerably to increase the readability.

Major comment

The clear first objective of this paper is to characterize the prevalence and persistence of symptoms up to 18 months after diagnosis in COVID-19 patients, which will interest many readers. In the abstract, the methods, results (apart from the last sentence) and conclusion address this question, and these sections in the abstract are clearly written.

However, at the end of the introduction (line 65-67) an additional second objective is introduced, concerning  the association between various tests (function tests, imaging),  physical performance and self-rated health status,  and persistent symptoms at medium term. This seems to be added to the paper in a later stage, as the introduction does not lead up to this objective (e.g. why would you investigate this). The statistical section concerning this second objective is very confusing and the results are not clearly presented either. I recommend the authors to keep the paper focused on the first objective and delete the second objective. This will highly improve the readability of this paper. They might consider writing another separate paper on the second objective.

Other comments

Abstract

1.      It is somewhat unclear how many timepoints were assessed. Are there two (3 and 18 months) or three (0 (baseline/discharge/diagnosis?), 3, and 18 months)?

2.     ‘Recovery’ is a peculiar term if participants are still suffering from on average 1.8 or 0.6 symptoms.

3.     I suggest to delete line 29-31 (last sentence of results). See also major comment.

Introduction

1.      The rationale for assessing “cardiopulmonary function tests, pulmonary imaging, physical performance, self-rated health status, and acute phase parameters” (only at medium term) is not clear. I would delete the second objective. See also major comment.

2.     In the second objective also acute phase parameters are mentioned. However, nowhere in the results have these been shown.

Methods

1.      Were consecutive patients included?

2.     When was a written informed consent obtained?

3.     Add to the primary outcome the time of assessment (line 127)

4.     Delete the secondary outcomes (line 129-130) as this is phrased as an association, not an outcome parameter. Furthermore, this does not fit with the second objective (line 67).

Statistical Analyses

In this paper many statistical analyses have been used, which are not needed or are even inappropriate. It only makes the paper unnecessary complex. Only descriptive statistics (such as mean, median, percentages, and so on) are needed to answer the first objective.

1.      Line 135-138. Unclear what the goal is of calculating pairwise correlations. The presentation of mean number of 8 symptoms suffices.

2.     Line 138-140. The objective of this paper was not to compare variables at acute, 3 and 18 months, but simply to describe the presence of these symptoms over time. No univariate statistical tests needed.

3.     Line 140. McNemar test for qualitative data does not make sense.

4.     Line 142-145. Why use factor analysis?

5.     Line 146-150. Is rather ‘technical’ information on how to perform statistical analyses.

6.     Line 150-158. Can be deleted if the second objective is deleted (see above). Furthermore, performing logistic regression on a continuous outcome (number of symptoms) does not make sense. It would be more interesting to know which (modifiable) variables measured at baseline would increase the risk of e.g. dyspnoea (yes/no) or weakness (yes/no) (these are still highly prevalent at 18 months).

Results

1.      Table 2. Many details are giving eg chest radiology. Would these be more suitable for an addendum?

2.     Table 2. Are median (min-max, or 25-75th percentile) more appropriate than mean (SD) for 6 MWD?  36-Short item form – are these values Mean (SD)?  Line 200 – lowest score for vitality scale (?).

3.     Line 204-219. Giving the mean number of symptoms at baseline, 3 and 18 months in one sentence suffices. The mean number is not that interesting. Much  more interesting, as the authors presented as well in figure 5, is the fluctuation of type of symptom over time. Describe these in the text. Delete lines 206-219. Figure 2, figure 3 (what is the purpose?) and figure 4 can be deleted.

4.     Line 252-300. Delete this part as this is very confusing. See major comment above. To clearly present these results tables are needed and the authors should think about what the dependent variable should be (e.g. dyspnoea / weakness/ cough?) in the main paper, other tables in addendum). Only presenting p-values does not give any information, nor does the presentation of correlation coefficients. Give odds ratios with 95%CI.

Discussion

A large part of the discussion focuses on the first objective (time course) and is clearly written. I suggest to delete those parts on prediction (eg parts from line 384 onwards).

1.      L308: equal proportion of men and women: was this to be expected? More recent studies have found a tendency of men towards higher severity and mortality (see e.g. Alwani et al., 2021; doi: 10.1002/rmv.2223).

2.     L373-374: in an attempt to move somewhat from speculation; apart from environmental aspects, could these also be explained by individual and interpersonal (socio-)psychological factors, such as anxiety, uncertainty, depression/depressed moods, inability to participate in society (work), stigmatisation etc., as discussed by many qualitative studies (e.g. Ladds et al. 2020, Kingstone et al. 2020, Buttery et al. 2021, Schaap et al. 2022)?

3.     L410: can you assess the health change scale usually included in the SF-36 as a (very) rough indicator?

Minor comments

Line 17 : n-55 should be n=55

Line 35: “pos-acute” should be post-acute

Line 96-98 The abbreviations PF, RF(??), BP and so on have not been used in the paper.

Line 166- 169. In Figure 1 (nice flowchart), it is not clear (not visible) why all 24 patients were excluded in the left arm.

Table 1. Present this more clearly and identically for each variable. Thus, no center alignment. Cases n(%) – put these in the column after eg HFNC n(%). Etc.

Line 196. 23.6%  (not 22.6)

Line 180-181 BIPA NIMV?

Line 191 It is not really laboratory examination. Rephrase.

Line 237: 21.7% for ageusia?

Overall: a check for extra or lack of spaces between words is necessary.

Author Response

We would like to thank the reviewers for their time and commitment in reviewing our manuscript. We thank the reviewers for their comments, which we believe were constructive and resulted in significant improvements to the manuscript. To address their concerns, we have answered all questions and modified the manuscript in several areas, as indicated in the point-by-point response below. 

Reviewer 2 Report

Dear authors,

You sought to provide insights into an emeging clinical entity, such as the Long COVID, and characterize the prevalence and the type of symptoms 3 to 18 months after acute infection. Although the submitted manuscript is well-written, there are several issues.

1. Did you record the rate of re-infections within the study population? It is reported that re-infection in patients with Long COVID may prolong or worsen the symptomatology. 

2. Did you assess the vaccination status of the study population during the follow-up period? Is there any chance that post-infection vaccination may have decreased the severity or the prevalence of the symptoms?

3. The study population was managed with treatment regimens that were available at the time of recruitment, but are not anymore indicated for COVID-19 (e.g. Hydroxychloroquine). Moreover, only 9% of the study population received the antiviral drug Remdesivir, which is part of the SOC treatment for hospitalized COVID-19 patients. Is there any possibility that the study population is not representive, thus impeding the interpretation of the results?

Minor issues

1. Line 184: High-flow nasal cannula is not NIVM. It should be corrected.

2.  Based on journal's recommendations and in compliance with the "instructions to authors", the abstract should be limited to 200 words and the headings should be removed.

3. A better design of the tables and the figure is essential. 

4. All the references should be revised and be described as indicated by the journal.

5. The list of references should be enriched.

Author Response

We would like to thank the reviewers for their time and commitment in reviewing our manuscript. We thank the reviewers for their comments, which we believe were constructive and resulted in significant improvements to the manuscript. To address their concerns, we have answered all questions and modified the manuscript in several areas, as indicated in the point-by-point response below. As recommended by Reviewer 1, we have rewritten and shortened the manuscript to improve readability. We have also removed the second objective regarding PFT's and echocardiography and the SF-36 questionnaire. Because of this change, several questions became irrelevant.

Round 2

Reviewer 2 Report

Thank the authors for the revisions they made.